# Integrative Application of Transcriptomics and Metabolomics Provides Insights into Unsynchronized Growth in Sea Cucumber (*Stichopus monotuberculatus*)

**DOI:** 10.3390/ijms232415478

**Published:** 2022-12-07

**Authors:** Bo Ma, Yang Liu, Wenjie Pan, Zhuobo Li, Chunhua Ren, Chaoqun Hu, Peng Luo

**Affiliations:** 1CAS Key Laboratory of Tropical Marine Bioresources and Ecology (LMB), Guangdong Provincial Key Laboratory of Applied Marine Biology (LAMB), South China Sea Institute of Oceanology, Chinese Academy of Sciences, Guangzhou 510000, China; 2University of Chinese Academy of Sciences, Beijing 100049, China; 3Southern Marine Science and Engineering Guangdong Laboratory (Guangzhou), Guangzhou 510000, China

**Keywords:** *Stichopus monotuberculatus*, unsynchronized growth, metabolomics, transcriptomics

## Abstract

Ever-increasing consumer demand for sea cucumbers mainly leads to huge damage to wild sea cucumber resources, including *Stichopus monotuberculatus*, which in turn exerts negative impacts on marine environments due to the lack of ecological functions performed by sea cucumbers. Aquaculture of sea cucumbers is an effective way to meet consumer demand and restore their resources. Unsynchronous growth is a prominent problem in the aquaculture of sea cucumbers which has concealed unelucidated molecular mechanisms until now. In this study, we carried out an integrative analysis of transcriptomics and metabolomics on fast-growing (SMF) and slow-growing (SMS) groups of *S. monotuberculatus* cultured in the same environmental conditions. The results revealed that a total of 2054 significantly differentially expressed genes (DEGs) were identified, which are mainly involved in fat digestion and absorption, histidine metabolism, arachidonic acid metabolism, and glutathione metabolism. 368 differential metabolites (DMs) were screened out between the SMF group and the SMS group; these metabolites are mainly involved in glycerophospholipid metabolism, purine metabolism, biosynthesis of unsaturated fatty acids, pyrimidine metabolism, arachidonic acid metabolism, and other metabolic pathways. The integrative analysis of transcriptomics and metabolomics of *S. monotuberculatus* suggested that the SMF group had a higher capacity for lipid metabolism and protein synthesis, and had a more frequent occurrence of apoptosis events, which are likely to be related to coping with environmental stresses. The results of this study provide potential values for the aquaculture of sea cucumbers which may promote their resource enhancement.

## 1. Introduction

Sea cucumbers are a kind of marine invertebrate that are widely distributed in oceans in the world [1,2]. Sediment-feeding sea cucumbers are often found in coral reef areas [3], which host a very important coral reef ecosystem, one of the most biodiverse ecosystems in the world [4]. The coral reef ecosystem has important ecological functions, such as providing a habitat for a wide range of marine life and protecting the coast from currents and waves [5]. Most sea cucumbers feed on organic debris in the marine sediments of the coral reef floor [3,6], and they are well known as environmental scavengers in marine ecosystems [7,8]. The feeding process of sea cucumbers effectively promotes the recycling of organic matter and nutrient salts from the substrate; after digestion and absorption, sea cucumbers excrete inorganic nitrogen and phosphorus. This form of nutrient cycling is vital to the coral reef ecosystem [7]. In recent years, ever-increasing consumer demand for sea cucumbers has directly resulted in the overfishing of wild sea cucumbers, and it has been reported that nearly 70% of resources of wild sea cucumbers have endured overfishing [9,10], which has led to a serious shortage of wild sea cucumbers, including *Stichopus monotuberculatus* [11]. As a result, the decline of wild sea cucumber resources may therefore cause a disturbance in the balance of the marine ecosystem. In China, wild resources of *S. monotuberculatus* are on the verge of depletion [12], which generates a threat to the health of marine ecosystem [13]. Therefore, artificial reproduction, releasing, and farming of *S. monotuberculatus* has become to be a key measure to restore sea cucumber resources and maintain the marine ecosystem.

Unsynchronous growth has always been a challenge in the rearing of aquatic animals [14]; it may cause a lot of problems, such as inconvenient management, longer farming cycles and the decrease in commercial value due to uneven sizes [15]. Previous studies have revealed that growth is a complex process, involving multiple regulatory effects of a large number of genes and metabolites [16]. Rapid advances in molecular and sequencing techniques over the last decade facilitate the large-scale identification of candidate genes associated with different phenotypes [17,18,19,20]. Transcriptomics can provide important information about gene expression levels that have a direct impact on phenotypes [21,22]. Metabolomics can provide intuitive data on tissue phenotypes in specific environmental conditions at specific time points [23]. Thus, combined transcriptomic and metabolomic analyses will provide a powerful method to understand key genes and metabolism mechanisms associated with interesting traits in animals or plants. As for sea cucumbers, transcriptomics and metabolomics on the growth performance mainly focused on *Apostichopus japonicus*, the most farmed species [24,25,26]. In tropical sea cucumbers, growth performance-related analyses have only been carried out in *Holothuria scabra* using transcriptome sequencing techniques [27]. Similar to other sea cucumbers, unsynchronous growth is evident in *S. monotuberculatus*; the molecular mechanisms behind its unsynchronous growth has stayed unknown until now. This study is the first to integrate transcriptomic and metabolomic analyses of tropical sea cucumbers of *S. monotuberculatus* to investigate the molecular mechanisms underlying their asynchronous growth.

In the present study, we used a metabolomic approach based on liquid chromatography–tandem mass spectrometry (LC–MS/MS) combined with transcriptomic analysis to investigate differences referring to metabolites and gene expression between the fast-growing (SMF) group and the slow-growing (SMS) group of *S. monotuberculatus*. Linkage networks were mapped based on correlations between metabolites and regulatory genes associated with the differences in growth performance. The results of this study will provide new insights into the molecular mechanisms of unsynchronous growth of sea cucumbers and a theoretical foundation for the artificial culture of sea cucumbers and the subsequent restoration of their biological resources.

## 2. Results

### 2.1. Significant Growth Performance Differences between SMF and SMS

After 120-d rearing, *S. monotuberculatus* showed significant differences in growth performance. The individuals of *S. monotuberculatus* were divided into the SMF group (3.32 ± 0.93 g) and SMS group (0.55 ± 0.09 g) according to their body weight, and the SMF group had much higher average weight than the SMS group (*p* < 0.001) (Figure 1 and Appendix A).

### 2.2. Metabolomics Analysis

To assess metabolomic differences between the SMF group and the SMS group, principal component analysis (PCA) and partial least squares discriminant analysis (PLS-DA) were performed based on LC-MS/MS metabolomic data. In positive and negative ion modes, the PCA score plots are shown in Figure 2A,B. The differential metabolites (DM) between the SMF groups and the SMS group were not well distinguished in the PCA model, which may be due to the limitations of the PCA model [28]. Compared with the PCA model, PLS-DA is a supervised model that reduces system noise and extracts variable information [29]. Therefore, the PLS-DA model has stronger classification capabilities than the PCA model [30]. Significant separation occurred between the SMF group and the SMS group in the PLS-DA scoring plot (Figure 2C,D), which clearly indicated that though sea cucumbers in the SMF group and the SMS group were grown in the same environment, the metabolite profiles of the two groups differed significantly. The quality of the PLS-DA model can be confirmed by the R2Y and Q2 values [31,32]. R2Y represents the contribution of the supervised model and Q2 represents the predictability of the PLS-DA model. Generally, a robust model has a Q2 > 0.4 [33]. In our results, the values of R2 and Q2 were 0.93 and 0.79 in the positive ion model, and 0.98 and 0.73 for R2 and Q2 in the negative ion model. The PLS-DA permutation test yielded a Q2 less than 0 in both ion modes (Figure 2E,F). These indicated that the PLS-DA model was not over-fitted, and was highly predictable and suitable for subsequent data analysis.

In total, 368 DMs were identified in the SMF group and the SMS group. Appendix A shows the details of the DMs. Among these DMs, 52 were yielded at higher concentrations in the SMF group than in the SMS group, and 316 metabolites were down-regulated in the SMF group compared with the SMS group. To explore the differences in metabolic pathways between the SMF group and the SMS group, these DMs were assigned to metabolic pathways using the Kyoto Encyclopedia of Genes and Genomes (KEGG) pathway analysis. This analysis revealed the differences in metabolic pathways between the SMF group and the SMS group and indicated that the pathways, such as metabolic pathways, pyrimidine metabolism, purine metabolism, regulation of lipolysis in adipocytes, and ABC transporters are very likely involved in the growth regulation of *S. monotuberculatus* (Figure 3A and Appendix A). To show the relative levels of DMs between the two groups, we performed a hierarchical cluster analysis (Figure 3B). The results showed that the metabolites differed significantly between the SMF group and SMS group.

### 2.3. Transcriptomic Analysis

#### 2.3.1. Transcriptomic Data Sequencing and Quality Control

A total of 25.06 million and 25.24 million high-quality sequence data were obtained for the SMF group and the SMS group, respectively. The mean GC content of the SMF group and the SMS group was 43.21% and 42.45%, respectively. Q20 and Q30 of the SMF group and the SMS group were 97.57% and 97.65%, and 92.88%, and 93.03%, respectively (Appendix A). The above data indicated a high quality of transcriptome sequencing, which laid a solid foundation for subsequent transcriptomic analysis. Clean data sets were deposited in the NCBI Sequence Read Archive (SRA) under the accession numbers PRJNA880726.

#### 2.3.2. Identification of DEGs between the SMF and the SMS Group

*p*-Vaule < 0.05 and |log2 FC| > 1 were used to identify differentially expressed genes (DEGs) in the SMF group and the SMS group, and a total of 2054 significant DEGs were identified in this way and shown in Appendix A. Compared with the SMS group, there were 969 up-regulated genes and 1085 down-regulated genes in the SMF group. The volcano map (Figure 4A) showed the distribution of DEGs between the two groups. A hierarchical clustering analysis of DEGs was performed and the results were shown in Figure 4B. From the figure, it could be seen that differentiated patterns of gene expression occurred between the SMF group and the SMS group.

#### 2.3.3. GO and KEGG Enrichment Analysis of DEGs

To further analyze the DEGs, we conducted a gene ontology (GO) enrichment analysis (Figure 4C and Appendix A). These DEGs were classified into three categories: biological processes, molecular functions, and cellular components. The main terms in biological processes were oxidation–reduction process, proteolysis, and G-protein coupled receptor signaling pathway. Protein binding, calcium ion binding, and metal ion binding are the main terms in molecular function. Among the cellular components, integral components of membranes, cytoplasm, and membranes were the main terms. To explore the differences in metabolic pathways between the two groups, we mapped DEGs to the KEGG pathway. Figure 4D and Appendix A show the top 20 metabolic pathways that were significantly enriched. Of these 20 pathways, Parkinson’s disease, thermogenesis, and phagosome were the three most enriched pathways. In addition, glycolysis/gluconeogenesis, fat digestion and absorption, and arachidonic acid metabolism pathways were also enriched.

#### 2.3.4. Quantitative Real-Time PCR Validation of the DEGs

To confirm the reliability of the transcriptome sequencing results, we randomly selected 11 DEGs for quantitative real-time PCR (qRT-PCR) validation. They included six up-regulated genes (*Rdh7*, *Pde10a*, *hnmt*, *PLB1*, *DHRS4*, and *GABBR2*) and five down-regulated genes (*NMT1*, *mt-Co3*, *SGPL1*, *Wnt7B*, and *WNT-1*) in the SMF group. The results showed that the qRT-PCR gene expression patterns were generally consistent with the RNA-seq results (Figure 4E), which demonstrated the reliability and accuracy of RNA-seq.

### 2.4. Integrative Analysis of Metabolomics and Transcriptomics

Spearman calculations were used to show the correlation between transcriptomic and metabolomics data. DEGs and DMs between the SMF and SMS groups are shown in Figure 5, and it exhibited a strong correlation between transcripts and metabolites (Appendix A).

Analysis of DEGs and DMs between the two groups using the KEGG pathway (Figure 6) shows that the biosynthesis of unsaturated fatty acids, purine metabolism, pyrimidine metabolism, arachidonic acid metabolism, glycerophospholipid metabolism, taurine and hypotaurine metabolism, apoptosis signal pathway and sphingolipid metabolism likely influenced the growth performance of *S. monotuberculatus*. Unsaturated fatty acids such as DHA and EPA were significantly down-regulated, and *Acox3* and *FABP3* expression were up-regulated in the SMF group. In the SMF group, the abundance of guanosine, adenosine, thymine, thymidine, uridine, and cytosine was lower, the abundance of uric acid was higher, and the expression of *Nt5e* was up-regulated. These results indicated that the SMF group of *S. monotuberculatus* may consume more energy and have a higher capacity for lipid metabolism and protein synthesis. In addition, we found lower accumulation of ARA, lysoPC and taurine and a lower expression of *PLA2* in the SMF group, which implied a higher resistance to oxidative stress in the SMF group. We also noted that in the SMF group, sphingosine was depleted, and genes associated with apoptosis such as *CTSD* and *cytC* were significantly up-regulated.

## 3. Discussion

### 3.1. The SMF Group of S. monotuberculatus Possesses a Strong Capacity for Lipid Metabolism and Protein Synthesis

Highly unsaturated fatty acids (HUFAs) such as docosahexaenoic acid (DHA, 22:6, n-3) and eicosapentaenoic acid (EPA, 20:5, n-3) have a positive role in regulating gene expression and lipid metabolism [34]. It has been well confirmed that HUFAs can significantly affect fundamental membrane properties such as acyl chain order, fluidity, elastic compressibility, phase behavior and permeability [35,36] and play a beneficial role in stabilizing dynamic membranes [37], membrane organization and cell division [38]. Metabolomic analysis in the present study showed significantly lower levels of DHA and EPA in the SMF group compared with the SMS group, which is consistent with the result found in ridgetail white prawn (*Exopalaemon carinicauda*) [39] and Pacific white shrimp (*Litopenaeus vannamei*) [40]. Acyl-CoA Oxidase (Acox) is a key enzyme involved in the β-oxidation of fatty acids [41]. The expression of Acox was found to be up-regulated in transcriptomic analysis in the SMF group. Fatty acid binding proteins (FABPs) belong to the lipid binding protein superfamily and play a role in intracellular fatty acid transport [42,43], of which FABP3 is mainly involved in the uptake and metabolism of fatty acids in muscle [44,45]. In this study, a more intense up-regulation of lipid metabolism occurred in the SMF group; notably, *FABP3* was found to be highly up-regulated in the SMF group. Based on these results and analysis, we speculated that the fast growth of *S. monotuberculatus* requires more active lipid metabolism, and thus HUFAs are largely consumed during the fast growth of the sea cucumber. It probably resulted in a lower level of EPA and DHA in the SMF group if they could not be supplemented in a timely manner. Previous studies have confirmed that appropriate amounts of DHA and EPA in diets can improve the growth rate, feed conversion rate and survival rate of aquatic animals [46,47,48]. In this respect, we considered that EPA and DHA should be moderately added to the diet of sea cucumbers to ensure the normal growth of sea cucumbers.

In the SMF group, the abundance of guanosine, adenosine, thymine, thymidine, uridine, and cytosine was all lower, however, the levels of uric acid were higher. In addition, there was a relatively high level of expression of 5-nucleotidase (*Nt5e*), an enzyme capable of catalyzing the hydrolysis of nucleotides by releasing phosphate from the 5-position of the pentose ring [49]. Purines and pyrimidines are the products of nucleotide catabolism and can be taken up by cells in other organs for protein synthesis [50]. Purine and pyrimidine nucleotides are major energy carriers, subunits of nucleic acids and precursors for the synthesis of nucleotide cofactors such as NAD and SAM [51]. The importance of purine and pyrimidine metabolism for the fast growth of Chinese mitten crab (*Eriocheir sinensis*) has been recognized [52]. Research on dietary nucleotides in fishes have shown they can improve growth in the early stages of development and enhance larval quality [53]. At a very earlier age, it has been found that the rate of nucleic acid synthesis is also higher in growing cells than in confluent cells [54]. In this work, nucleotide metabolism was more active and the levels of purine and pyrimidine were lower in the SMF group than in the SMS group. Therefore, it is reasonable to speculate that purines and pyrimidines are intensively utilized to support the faster growth and development of *S. monotuberculatus*, and this result may also imply an important application of nucleotides in the diets of sea cucumbers.

### 3.2. The SMS Group Experienced Stronger Oxidative Stress

Arachidonic acid (ARA) is an n-6 long-chain polyunsaturated fatty acid (LC-PUFA) that is an essential fatty acid for marine fishes [55,56]. Metabolomic data showed that free ARA levels were significantly higher in the SMS group than in the SMF group. Normally, ARA is not present in free form, but as phospholipids in the cell membrane. When the cell membrane encountered stimulation, phospholipase A2 (PLA2) was activated, causing ARA to be released from the cell membrane [57,58]. Transcriptomic data showed that PLA2 expression was up-regulated in the SMS group, which was consistent with increased levels of arachidonic acid in the SMS group. It has been shown that high levels of n-6 series fatty acids increase the expression of inflammatory factors (IL-1β and TNFα) in fishes [59]. Excessive inflammatory responses in the body can lead to oxidative stress in animals [60]. It has been confirmed that arachidonic acid metabolism pathway is closely linked to stress processes and plays an important role in signal transduction pathways as well [61,62,63,64]. Arachidonic acid metabolism was also significantly enriched in unsynchronously growing red swamp crayfish (Procambarus clarkii) [65].

On the other hand, we found significantly higher levels of LysoPC in the SMS group than in the SMF group. Lecithin (PC), cephalexin (PE) and phosphatidylserine (PS) are all phosphoglycerides, which are important components of cell membranes and are the basis of cellular metabolism, energy metabolism and signal transmission [39]. Lysophospholipids are a class of biologically active signaling lipids produced by phospholipase-mediated hydrolysis of membrane phospholipids (PLs) and sphingolipids (SLs) [66]. PLA2 can produce lysoPCs by hydrolyzing oxidatively truncated PCs (oxPCs) during low density lipoprotein oxidation [67,68]. It has been shown that over expression of LysoPC is associated with the occurrence of high levels of oxidative stress in the organism [68,69].

Taurine is a β-sulfur amino acid that may act as an antioxidant in living organisms [70,71]. In different experimental systems, taurine has been shown to act as a direct antioxidant with a scavenging effect on oxygen-free radicals and as an antioxidant to prevent changes in ion transport and membrane permeability due to oxidant damage [72]. In fish, taurine has been shown to reduce lipid peroxidation and oxidative stress levels in red seabream (Pagrus major) and European seabass (Dicentrarchus labrax) [73,74]. The free taurine content was significantly higher in the SMF group than in the SMS group, which therefore suggested that a lower level of free taurine in the individuals of the SMS group may affect their ability to counter oxidative stress.

By combining these changes of metabolites related to the response to oxidative stress, we presume that under a constantly changing environment, the SMS group may be more sensitive to environmental changes and may lack a perfect mechanism of metabolism and transcription to respond to oxidative stress; thus, they must use more energy to cope with these stresses, at the expense of normal growth.

### 3.3. Apoptosis Occurred More Frequently in the SMF Group

Apoptosis is a process of programmed cell death that is regulated by gene expression. Throughout the life cycle of multicellular organisms, apoptosis is responsible for tissue remodeling and normal organization [75]. Apoptotic pathways mainly include the extrinsic apoptotic pathway and the mitochondrial apoptotic pathway [76]. The external pathway is regulated by extracellular signals, whereas the mitochondrial apoptotic pathway is triggered by a complex including cytochrome C [77]. Sphingolipids ensure the structural integrity of the cell membrane, and regulate cellular processes such as apoptosis, proliferation, and senescence [78]. Metabolites of sphingolipids, such as sphingosine, may act as a stimulus to enhance the permeability of the lysosomal membrane, leading to the release of cathepsin D from the lysosome into the cytoplasm [79,80]. CTSD belongs to the pepsin family of aspartic proteases and is an important member of the cathepsin family [81,82], which can lead to the release of CYTC from mitochondria by cleaving proteins [80]. CYTC in turn activates downstream caspase 3 and caspase 9, thereby inducing apoptosis [83]. Caspases are a class of proteases that play an important role in the induction and execution of apoptosis. When caspases are activated, they acquire the function of cleaving substrates, leading to the biochemical and morphological changes associated with apoptosis in cells [84]. Combining transcriptomic and metabolomic data, it could be seen that the accumulation of sphingosines was reduced in the SMF group and the expression of genes associated with apoptosis such as CTSD and cytC were significantly up-regulated, indicating that more apoptotic events may have occurred in the SMF group. In swimming crab (*Portunus trituberculatus*), similar up-regulation of genes in the apoptotic pathway was found in the SMF group [85]. Based on the aforementioned analysis that individuals in the SMS group may be more sensitive to environmental stresses, we tentatively speculated that the sensitivity to environmental stresses likely decreases the level of apoptosis in the SMS group. A similar phenomenon was also observed—that genes in apoptotic pathway were down-regulated—when rainbow trout were farmed at high densities (representing environmental stress). Further studies on the relationship between apoptosis and growth in sea cucumbers should be carried out in the future.

## 4. Materials and Methods

### 4.1. Experimental Animals

Parental sea cucumbers collected from Sanya West Island (N 18°24′, E 109°38′), Hainan, China, were artificially stimulated using the Dry Stimulation method [86]. Fertilized eggs were transferred to concrete culture ponds. Following the method of Chen et al., they were fed 1 g of chlorella powder and 1 g of yeast powder daily. After 15 days, they were fed 1 g chlorella powder, 2 g sargassum powder, 1 g bacillus, 1 g yeast and 3 g sea cucumber feed supplement [28]. After 120-day culture, *S. monotuberculatus* seedlings of significantly different sizes were obtained and prepared for sampling. The use of animals in this study was approved by the Animal Research and Ethics Committees of the South China Sea Institute of Oceanology, Chinese Academy of Sciences. All experiments were conducted following the guidelines of the committees.

### 4.2. Analysis of Growth Characters and Sampling

One hundred and fifty seedlings of *S. monotuberculatus* were randomly collected from a pool and weighed using an electronic balance (0.01 g accuracy). Each SMF group contained 18 seedlings with large sizes, and likewise, each SMS group contained 18 seedlings with small sizes. Body wall tissue from each sea cucumber was taken on ice, placed in enzyme-free lyophilisation tubes, quickly frozen in liquid nitrogen and then transported to the laboratory in dry ice. Body wall tissues from three individuals of the same group were mixed as one sample (one biological replicate), and thus six samples were obtained for the SMF group and the SMS groups, respectively.

### 4.3. LC-MS/MS Analysis Conditions

Six samples from the SMF group and six samples from the SMS group were used for non-targeted metabolomics assays. 50 mg of each sample was added into an EP tube filled with 500 µL of pre-cooled 80% methanol and was homogenized followed by the centrifugation for 10 min at 20,000× *g*. The supernatant was transferred to a new 1.5-mL centrifuge tube and dried in freeze. Each dried extract was re-dissolved with 100 μL of pre-cooled 80% methanol and an equal part of each sample (10 μL) was taken as a polled QC sample. LC-MS was performed using an UltiMate 3000 HPLC system (Thermo, Waltham, MA, USA). Reverse phase separations were performed using an ACQUITY UPLC T3 column (100 mm*2.1 mm, 1.8 µm, Waters, Devon, UK). The column oven temperature was maintained at 50 °C. The flow rate was 0.3 mL/min and the mobile phase consisted of phase A (water, 0.1% formic acid) and phase B (acetonitrile, 0.1% formic acid). The gradient elution conditions were set as follows: 0 to 0.8 min, 2% B; 0.8 to 2.8 min, 2% to 70% B; 2.8 to 5.6 min, 70% to 90% B; 5.6 to 8 min, 90% to 100% B; 8 to 8.1 min, 100% to 2% B; 8.1 to 10 min, 2% B. The injection volume for each sample was 4 µL.

Mass spectral data were collected using a Thermo UHPLC-Q Exactive Mass Spectrometer equipped with an electrospray ionisation (ESI) source in positive and negative ion modes. Ion source (ESI) parameters were set as follows: spray voltage (|KV|)) to 4000 in positive ion mode and to 4000 in negative ion mode; sheath gas flow rate to 35; aux gas flow rate to 10; and capillary temperature to 320 °C. The injection volume for each sample was 4 µL. Two wash samples were scanned first, followed by 3–4 QCs; then, one QC was inserted for every 10 scanned samples.

### 4.4. Metabolomics Data Analysis

The acquired raw mass spectrometry data (.raw files) were imported into Compound Discoverer 3.1.0 (Thermo, Waltham, MA, USA) for data pre-processing, including peak extraction, intra- and inter-group retention time correction, adduct ion merging, gap filling, background peak labeling, and metabolite identification. Each ion was identified by combining retention time and *m*/*z* data. The intensity of each peak was recorded and final information, such as characteristic molecular weight, retention time, peak area, and identification results was output. Using online KEGG and HMDB databases, metabolites were annotated by matching the exact molecular mass data, name, and formula of the samples to data in the databases. If the mass difference between the observed and database values was less than 10 ppm, the metabolite was annotated.

Analysis methods included PCA and PLS-DA analysis. A Student′s *t*-test was used to detect differences in metabolite abundance between the SMF group and the SMS group. Multiple adjustments for *p* values were performed using FDR (Benjamini-Hochberg). Supervised PLS-DA was performed by metaX, and VIP values were calculated to distinguish different variables between groups. Univariate analysis of fold-change and *p*-value from the Student′s *t*-test, combined with multivariate statistical analysis of VIP (variable important for the projection) from PLS-DA were used to screen for differential metabolites (DMs). The differential ions had to satisfy three criteria: (1) ratio ≥ 2 or ratio ≤ 0.5; (2) *p*-value ≤ 0.05; (3) VIP ≥ 1. MetaboAnalyst 5.0 (http://www.metaboanalyst.ca (accessed on 5 August 2022)) was used to analyze the DMs based on KEGG database.

### 4.5. Transcriptomics Analysis

Body wall tissues were used to extract RNAs for RNA sequencing (RNA-seq). RNAs were extracted using a Trizol reagent (Invitrogen, Carlsbad, CA, USA) following the manufacturer’s procedure. The quantity and purity of total RNAs were obtained by Bioanalyzer 2100, and RNA 1000 Nano LabChip Kit (Agilent, Santa Clara, CA, USA) with RIN number > 7.0. Poly (A) RNAs were purified from total RNA (5 µg), using poly-T oligo-attached magnetic beads, in a total of two rounds of purification. Next, the mRNAs were fragmented into small pieces with divalent cations at high temperature. The cleaved RNA fragments were then inverted according to the protocol of the RNA-Seq sample preparation kit (Illumina, San Diego, CA, USA) to generate the final cDNA library with an average insertion size of 300 bp (±50 bp) for the paired-end libraries. These libraries were then sequenced in paired-end on an Illumina Novaseq™ 6000 according to the vendor’s recommended protocol (LC Sciences, Hangzhou, Chian).

After RNA-seq, raw reads from the transcriptome dataset were filtered. Cutadapt [87] and Perl scripts were used to remove reads, including adaptor contamination, low-quality bases, and undetermined bases. Reads were then verified using FastQC (http://www.bioinformatics.babraham.ac.uk/projects/fastqc/ (accessed on 3 August 2022)) to verify sequence quality, including the Q20, Q30, and GC content of the clean data. All downstream analyses were performed based on high-quality clean data. De novo assembly of transcriptomes was performed using Trinity 2.4.0 [88]. Trinity groups transcripts into clusters based on shared sequence content, and such transcript clusters are loosely referred to as ‘genes’. The longest transcript in the cluster is selected as the ‘gene’ sequence (also known as Unigene). Expression levels of Unigenes were estimated using Transcript Per Million (TPM) [89] by Salmon [90]. Differentially expressed genes (DEGs) were selected with |log2 (fold change)| > 1 and with *p*-value < 0.05 by R package edgeR [91]. Further enrichment analysis of DEGs was performed using KEGG and GO databases.

### 4.6. qRT-PCR Validation

To verify the accuracy of the RNA-seq results, 11 DEGs were randomly selected for qRT-PCR. Primers for qRT-PCR were designed using primer premier 5. The primers used in this study were listed in Table 1. The relative expression values of DEGs were calculated using the 2^−ΔΔCt^ method [92].

### 4.7. Integrative Analysis of Metabolomics and Transcriptomics

DEGs (*p* value < 0.05, |log2 (fold change)| > 1) and DMs (*p* value < 0.05, |log2 (fold change)| > 1 and VIP > 1) between the SMF and the SMS group were used for integrative analysis. Pearson’s method was used to analyze the data correlation coefficients between metabolomics and transcriptomics by R package [93]. Heat maps were used to show the association between DEGs and DMs.

### 4.8. Growth Data Statistics

Student′s t-test was used to compare the difference in growth performance between the SMF and SMS groups. The significance levels in the analysis were considered at *p* < 0.05. Results were presented as mean ± standard deviation (S.D.). The analysis was carried out using SPSS 19.0 software.

## 5. Conclusions

Integrating transcriptomics and metabolomics provides novel insights into the molecular mechanisms underlying the unsynchronous growth of *S. monotuberculatus*. The analysis showed that there were significant differences in transcripts and metabolites between the SMF group and the SMS groups. The SMF group may have had a higher capacity for lipid metabolism, protein synthesis, energy metabolism, and underwent frequent apoptotic events. The SMS group may have experienced stronger oxidative stress. These likely cause the unsynchronous growth of *S. monotuberculatus*. This study is the first to integrate the transcriptome and metabolome to investigate the mechanism of unsynchronous growth in *S. monotuberculatus*. The results will provide a theoretical basis for better development of sea cucumber farming, through molecular breeding using the key genes mentioned in the paper, and also through the use of key metabolites such as DHA and EPA as feed additives for sea cucumbers. Ultimately, the recovery of sea cucumber resources will be promoted.

## Figures and Tables

**Figure 1 ijms-23-15478-f001:**
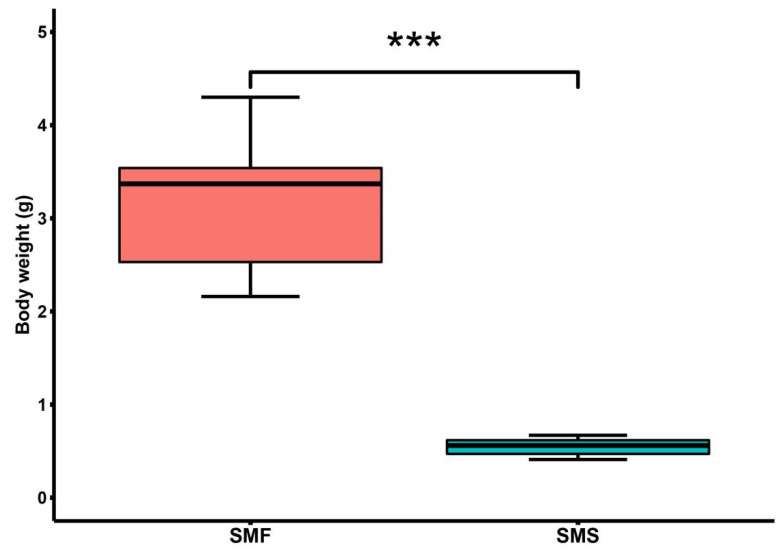
Comparison of the growth performance of *S. monotuberculatus* in the SMF group and the SMS group. “***” indicates *p* < 0.001.

**Figure 2 ijms-23-15478-f002:**
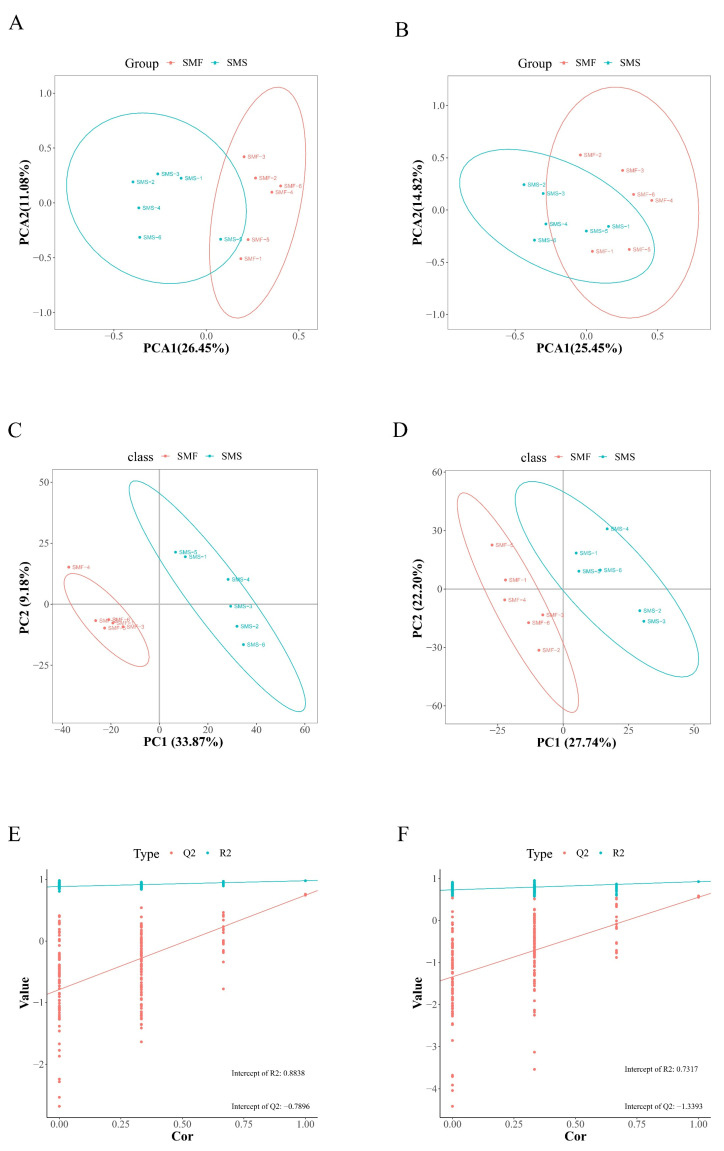
Quality analysis of metabolomics data. (**A**) The PCA scores plot of samples acquired in the positive ion mode. (**B**) The PCA scores plot of samples acquired in the negative ion mode. (**C**) The PLS-DA score plot for the positive ion mode. (**D**) The PLS-DA score plot for the negative ion mode. (**E**) The PLS-DA validation for the positive ion mode. (**F**) The PLS-DA validation for the negative ion mode.

**Figure 3 ijms-23-15478-f003:**
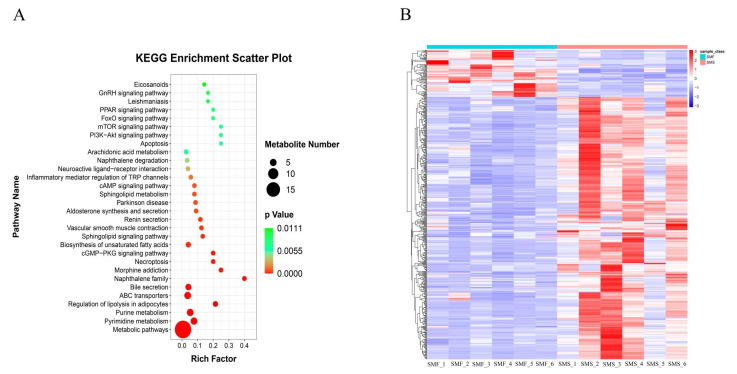
Hierarchical clustering analysis for DMs and the metabolomics view map of the significant metabolic pathways. (**A**) KEGG pathway analysis of metabolites differentially expressed between the SMF group and the SMS group. The vertical coordinates indicate the top 30 KEGG terms significantly enriched by DEMs and the horizontal coordinates indicate the enrichment factor between the two sampled datasets. (**B**) Hierarchical clustering analysis of DMs between SMF and SMS groups. Red indicates metabolites up-regulated in SMF; blue indicates metabolites up-regulated in SMS.

**Figure 4 ijms-23-15478-f004:**
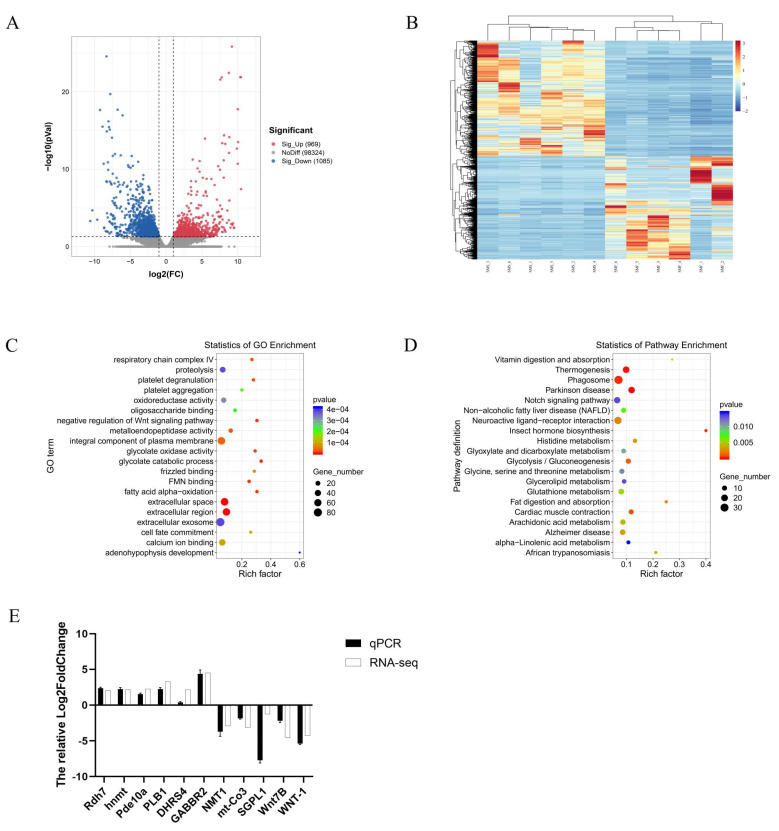
Transcriptome analysis of *S. monotuberculatus* in the SMF and SMS groups. (**A**) Volcano plot of DEGs between the SMF group and the SMS group. (**B**) The hierarchical clustering analysis of the DEGs between the two groups. (**C**) GO functions of DEGs between the two groups. (**D**) KEGG functions of DEGs between the two groups. (**E**) Comparison of gene expression data from RNA-Seq and qRT-PCR. Data are expressed as the mean ± standard deviation (SD) of three replicates.

**Figure 5 ijms-23-15478-f005:**
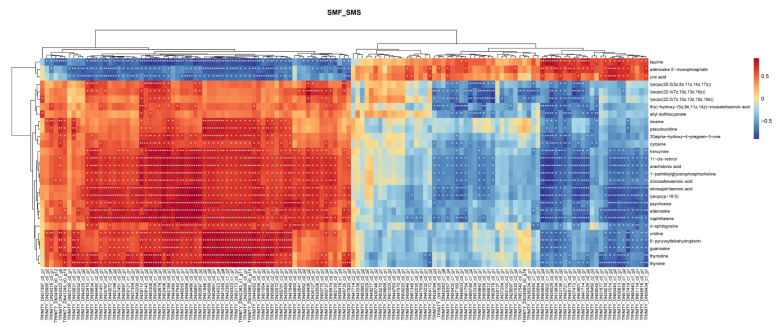
Heat map of the correlation between the transcriptome and metabolome of *S. monotuberculatus*. The columns are genes and the rows are metabolites; they show the relationship between genes and metabolites. Red indicates that genes and metabolites show positive correlation, blue indicates negative correlation between genes and metabolites. “*” indicates a significant correlation between genes and metabolites (*p* < 0.05). “**” indicates *p* < 0.1.

**Figure 6 ijms-23-15478-f006:**
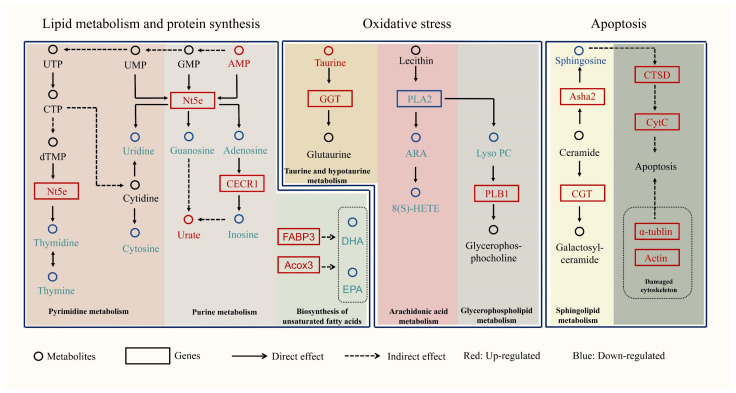
An integrative metabolic network map inferred from differential genes and metabolites between the SMF group and the SMS group identified by transcriptome and metabolome integration analysis. DEGs are indicated by boxes, among which red indicates up-regulated genes and blue indicates down-regulated genes in the SMF group. DMs are indicated by circles, among which red indicates up-regulated metabolites and blue indicates down-regulated metabolites in the SMF group. Abbreviations: UTP, Uridine-5’-triphosphate; CTP, Cytidine-5’-triphosphate; dTMP, deoxynucleotide; Nt5e, 5’-nucleotidase; UMP, uridylic acid; GMP, Guanosine monophosphate; AMP, Adenosine 5’-monophosphate; PLA2, phospholipase A2; ARA, Arachidonic acid; 8(S)-HETE, (5Z,9E,11Z,14Z)-(8S)-8-Hydroxyeicosa-5,9,11,14-tetraenoic acid; Lyso PC, 2-Lysolecithin; PLB1, lysophospholipase 1; GGT, gamma-glutamyltranspeptidase; FABP3, fatty acid-binding protein 3; DHA, Docosahexaenoic Acid; EPA, Eicosapentaenoic Acid; CECR1, adenosine deaminase; Asha 2, neutral ceramidase; CGT, ceramide galactosyltransferase; CTSD, cathepsin D; Cyt C, Cytochrome C; Acox3, Acyl-CoA Oxidase 3.

**Table 1 ijms-23-15478-t001:** The sequence information of primers used for qRT-PCR.

Gene Name	Primer (5′→3′)
*Rdh7*	F: ACTACGCAATACAGGTTGTT R: CCCATTGAAAGCTGCTTATG
*Pde10a*	F: GGTACCTGCTAACTTTGACA R: ATCAGGACCAAAAAGGTTGT
*hnmt*	F: TATATCCCCAGAAGGTCGTT R: TGGTTCTATGACTGTGTTGG
*PLB1*	F: GACTTCACCATTCTGAGGGA R: TAATCGCTCAGCCATGTACT
*DHRS4*	F: CACTGCCTACTTGTTACTGT R: GAGCTCAGTGTACGAATCAT
*GABBR2*	F: CGGACAATGGTTTATTGGCT R: TTAAACATTGGAGGGGACCA
*NMT1*	F: TATTTGGCGAAGGAGTTTGA R: TTACAGCCATCTCCTGTCTA
*mt-Co3*	F: TTTTCAGCCCTCCTTCTAAC R: TCCTTCACGAATTACGTCTC
*SGPL1*	F: CGACTGAGAAAGAACAAGGA R: GGAACTGCAAGGAGTTTAGA
*Wnt7B*	F: TGTGACAAAAGGTATCCCGA R: AGCATCTACGAACTCTCTGG
*WNT-1*	F: ACGAGCCTGTTCGGTAA R: CCCACTCCCAGTCTTCA
*β-actin*	F: GAGGTCTGCAATACCTGCGATT R: TTTTGTGTGGGGTGTGGTTG

## Data Availability

The data presented in this study are available in the article.

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
