# Peer review of "Integrative Application of Transcriptomics and Metabolomics Provides Insights into Unsynchronized Growth in Sea Cucumber (Stichopus monotuberculatus)"

_ijms, 2022, doi:10.3390/ijms232415478_

Round 1
Reviewer 1 Report
In this manuscript, the authors try to investigate the factors regulating the unsynchronous growth of the sea cucumber, S. monotuberculatus, through an integrative analysis of transcriptomics and metabolomics on fast-growing (SMF) and slow-growing (SMS) groups. The authors found that the SMF group had a higher capacity for lipid metabolism and protein synthesis and had more frequent occurrence of apoptosis events than SMS group. The work is technically of high quality and has several interesting/valuable findings, and it revealed potential genes and metabolisms that regulate the unsynchronized growth of sea cucumber.
Some major concerns and issues that need attention are as follows:
1. Are the larve and seedlings cultured in the same conditions, for SMF and SMS groups? Where did the environmental stresses come from for the SMS group during the culture?
2. As for the metabolomics analysis, PLS-DA scoring plot (Figure 2C and 2D) shows a significant separation between the SMF group and the SMS group, and there is no overlap between the 2 groups, whereas the PCA score plots shown in Figure 2A and 2B present an overlap between the 2 groups. Whether we can combine the 2 models and figure out the DMs shown in both models?
3. What does the metabolic pathway in line 128 and in Figure 3A refer to? Does it not contain the other metabolisms, such as in the Figure 3a, pyrimidine metabolism and purine metabolism?
4. Line 40-44:The authors emphasize the coral ecosystem at the beginning of the introduction, but the description of the interaction between sea cucumber and coral ecosystem is too simple, and should summarize and list how the two mutually beneficial symbiosis.
5. Line 50-52:The author needs to reorganize this paragraph, which is similar to the Line 70. Suggest changing “only one” to other expressions.
Some minor mistakes should be modified, which are listed as below:
1. Line 15: This sentence has a grammatical problem and needs to be revised, e.g. change “lead” to “leads” and remove the “the” before the huge
2. Line 23: Change “different” to “differentially”
3. Line 63-64: If possible, a reference should be added to this sentence
4. Line 86:I think the title is not clear enough and should focus more on the weight difference in the results.
5. Line 189:The font format needs to be consistent, other subheadings also have the same problem, please check them one by one.
6. Line 206: Where is Acox3 in Figure 6? The Acox3 should be added in the Figure 6?
7. Line 208 and 259: The cytosine in Figure 6 is upregulated, not down-regulated. The urate in Figure 6 is down-regulated.
8. Line 212: The taurine in Figure 6 is upregulated, not down-regulated.
9. Line 215: Conclusion is missing. “sphingosine was depleted and genes associated with apoptosis such as CTSD and cytC were significantly upregulated, indicating....”
10. Line 216: The border on this figure is too wide and it is recommended that the proportion of the light pink border be reduced.
11. Line 242-267: Please check the section carefully, some changes need to be made e.g. at line 242 “involving” should be changed to “involved”, at line 259 “were” should be changed to “was” and at line 267 “researches” should be changed to “research”.
12. Line 351-352:Duplicate references, please revise.
13. Line 357: Provide more information about ponds and farming methods.
14. Line 410: Please have a uniform format for the details of the reagents used in the study.
15. Line 448: Please provide the attribution for the R package used in the study.
16. In the conclusion, authors should be based on the current findings and research questions, summarize each of the points mentioned in the discussion.
Reviewer 2 Report
In this manuscript, the authors used an integrated approach of transcriptomic and metabolomic techniques to explore the potential molecular mechanisms underlying the unsynchronized growth of fast-growing and slow-growing sea cucumbers (Stichopus monotuberculatus) in the same growth environment. The results obtained from this work provide an important theoretical basis for the artificial culture of sea cucumbers. The article is well written, addresses a relevant problem. The methodology used is sound and the mechanism analysis is deep enough. I recommend it for publication, upon responding to a couple of comments/suggestions below.
1. There are many abbreviations in this article. It is suggested that the authors illustrate the abbreviations uniformly.
2. Please refine the "Conclusion" section to explain how the results provided in this work provide a basis for a better development of farming industry of sea cucumbers.
3. In line 174, the authors mentioned that "parkinson disease" is one of the most enriched metabolic pathways. However, this pathway is relevant to humans. The authors need to clarify whether this pathway is present in sea cucumbers. If not, this pathway should be excluded from the article and the related figures.
4. Some terms in Table S2 were written in Chinese, please revise them to English.
Reviewer 3 Report
This paper presents transcriptomics and metabolomics data, compared between fast-grown and slow-grown individuals of cultured seedlings of the seacucumber Stichopus monotuberculatus.
Knowledge has been very limited about transcriptomics and metabolomics of sea cucumbers, as explained in the introduction of this paper, and thus, I believe that the contents of this paper is valuable. Generally, the complex data were well analyzed, presented and explained. However, I question whether the transcriptomic and metabolomic differences between the two groups shown in this paper really represent those of unsynchronous growth. At the time of sampling, the two groups already showed very different body sizes (>3 times difference in body weight). Therefore, the differences possibly reflected just ontogenetic changes in transcriptomics and metabolomics specific to this species, not unsynchronity of growth among individuals. Before binding this result with the unsynchronous growth, the authors need to show the ontogenetic pattern of transcriptomics and metabolomics of this species and confirm their stability at the sampling timing. At least, samplings at the initial and final of the period during which these gropus showed different growth rates seem necessary. Otherwise, the authors need to discuss these bachgroud factors (effect from growth rate or body size) and clearly state both what can be concluded from the data and what remains unclear regarding the theme (=unsynchronized growth).
Please also add discussion about the novelty of this study compared to previous similar studies using other sea cucumbers.
Minor comments
L48, L51 Is there any reference about fishery or resource degradation of Stichopus monotuberculatus in China or other regions?
L94-96 Abbreviations should be explained here such as PCA, PLS-DA, and DM, although part of them are expained later in the methods section.
Figure 6 Very helpful figure to understand the results.
